# Role of Glutathione in Cancer: From Mechanisms to Therapies

**DOI:** 10.3390/biom10101429

**Published:** 2020-10-09

**Authors:** Luke Kennedy, Jagdeep K. Sandhu, Mary-Ellen Harper, Miroslava Cuperlovic-Culf

**Affiliations:** 1Department of Biochemistry, Microbiology and Immunology, University of Ottawa, 451 Smyth Road, Ottawa, ON K1H 8M5, Canada; lkenn012@uottawa.ca (L.K.); Jagdeep.Sandhu@nrc-cnrc.gc.ca (J.K.S.); mharper@uottawa.ca (M.-E.H.); 2Ottawa Institute of Systems Biology, University of Ottawa, 451 Smyth Road, Ottawa, ON K1H 8M5, Canada; 3Human Health Therapeutics Research Centre, Bldg M54, National Research Council Canada, 1200 Montreal Road, Ottawa, ON K1A 0R6, Canada; 4Digital Technologies Research Centre, Bldg M50, National Research Council Canada, 1200 Montreal Road, Ottawa, ON K1A 0R6, Canada

**Keywords:** glutathione, tumor therapy, tumor metabolism, nitrosoglutathione, *S*-nitrosation, ferroptosis, reactive oxygen species, metabolism modeling

## Abstract

Glutathione (GSH) is the most abundant non-protein thiol present at millimolar concentrations in mammalian tissues. As an important intracellular antioxidant, it acts as a regulator of cellular redox state protecting cells from damage caused by lipid peroxides, reactive oxygen and nitrogen species, and xenobiotics. Recent studies have highlighted the importance of GSH in key signal transduction reactions as a controller of cell differentiation, proliferation, apoptosis, ferroptosis and immune function. Molecular changes in the GSH antioxidant system and disturbances in GSH homeostasis have been implicated in tumor initiation, progression, and treatment response. Hence, GSH has both protective and pathogenic roles. Although in healthy cells it is crucial for the removal and detoxification of carcinogens, elevated GSH levels in tumor cells are associated with tumor progression and increased resistance to chemotherapeutic drugs. Recently, several novel therapies have been developed to target the GSH antioxidant system in tumors as a means for increased response and decreased drug resistance. In this comprehensive review we explore mechanisms of GSH functionalities and different therapeutic approaches that either target GSH directly, indirectly or use GSH-based prodrugs. Consideration is also given to the computational methods used to describe GSH related processes for in silico testing of treatment effects.

## 1. Introduction

Metabolic requirements of tumor cells are high due to their rapid proliferation rates. Tumor cell metabolism is also highly flexible owing to the capacity of cells to adapt to a range of microenvironment characteristics and cytotoxic therapies. With recent advances in metabolic research, there is an improved understanding of the complexities of tumor cell metabolism and redox chemistry in cancer etiology, progression, and treatment. There has also been an increased interest in the role of glutathione (GSH) metabolism, which is now known to extend beyond its well-known central role in the regulation of reactive oxygen species (ROS). While ROS are inevitable byproducts of oxidative metabolism, their production and neutralization are controlled by an extensive array of processes. ROS are inextricably linked to the cellular oxidation of fuels, when liberated electrons (reducing equivalents) modify the balance of reduced and oxidized pairs of electron acceptors such as NADH/NAD^+^ and NADPH/NADP^+^. During the complex and tightly regulated oxidation processes such as glycolysis in the cytosol, and citric acid cycle and oxidative phosphorylation reactions in the mitochondria, electrons are used to power the synthesis of ATP, which then supports the high energy demands of tumor cells. However, during fuel oxidation, electrons can escape and interact with oxygen to form ROS. Mitochondrial redox reactions are a major source of ROS in most cell types, including tumor cells. ROS are generated by the NADPH oxidase (NOX) enzymes, located on the plasma membrane of cells, notably in neutrophils and vascular endothelial cells. In order to avoid the toxic effects of ROS on DNA, proteins, and lipids, cells have evolved a wide array of antioxidant mechanisms, and this is particularly true in tumor cells where the activities of redox reactions are extremely high. Many of the ROS defense mechanisms depend on the tripeptide molecule, GSH. Tumors also have an increased burden of reactive nitrogen species (RNS) which are generated enzymatically by nitric oxide synthases (NOS), located in tumor cells, stromal cells or infiltrating immune cells. NOS activity produces NO gas that reacts non-enzymatically with cysteine residues in glutathione, major non-protein thiol and protein-thiols to form S-nitrosoglutathione (GSNO) and S-nitrosothiols in a process known as S-nitrosation. Through these reactions as well as its interactions with a number of proteins, glutathione is one of the major regulators of cancer cell development, progression, response to therapy and its environment leading to a number of avenues for therapeutic targeting (Figure 1).

In this review, we focus on the role of GSH in cancer cell metabolism, tumor microenvironment and communication leading to its function in tumor progression and treatment.

## 2. ROS Generation and Signaling

The major known forms of ROS include superoxide anion radical (O^−2^), hydrogen peroxide (H_2_O_2_) and/or hydroxyl radical (OH). Superoxide anion has a short half-life and is quickly converted to hydrogen peroxide by the superoxide dismutase enzymes, SOD1 and SOD2, located in the cytoplasm and mitochondria, respectively. In comparison to superoxide, hydrogen peroxide is much longer-lived (e.g., 1 ms vs. 1 us) [1,2], and can traverse membranes, making it capable of functioning as a signaling molecule. H_2_O_2_ can react with ferrous ions in a reaction known as the Fenton reaction to generate the short-lived hydroxyl radicals, which are highly toxic and can cause irreversible damage to essentially all types of cellular macromolecules, including nucleic acids (e.g., mutations), lipids (e.g., oxidized lipids such as 4-HNE), and proteins (e.g., protein carbonylation) [3,4,5]. Superoxide can also interact with nitric oxide (NO), resulting in the production of the reactive nitrogen species (RNS), peroxynitrite (ONOO^−^), which can cause nitration of protein tyrosine residues thereby affecting protein function [5].

In mitochondria the major sources of ROS include OXPHOS complexes I and III, and the dehydrogenases that use ubiquinone as an acceptor, and this includes succinate dehydrogenase (OXPHOS complex II). Other distinct mitochondrial sites associated with substrate oxidation and oxidative phosphorylation leak electrons to oxygen to produce superoxide or hydrogen peroxide, and these include oxoacid dehydrogenase complexes that shuttle electrons to NAD^+^ [6]. Mitochondrial membrane potential is also important in ROS production and cancer. It has long been known that a mild depolarization of the mitochondrial inner membrane inhibits mitochondrial ROS production [7]. This is one way through which the post-translational control of mitochondrial uncoupling proteins UCP2 and UCP3 by glutathionylation may decrease mitochondrial ROS production [8,9]. Recent evidence supports the idea that mild mitochondrial depolarization is crucial for the slowing of aging processes associated with oxidative damage [8].

Nonetheless, it is currently well recognized that low levels of ROS are important for many types of cell signaling reactions; this is especially true for the longer lived H_2_O_2_ [10]. One of the predominant mechanisms through which ROS mediate cell signaling processes is through the oxidation of protein cysteine thiols, thereby affecting protein structure and function. Low levels of H_2_O_2_ can oxidize these residues, leading to their sulfenylation (S-OH), sulfenamidylation (S-N), glutathionylation (S-SG), or disulfide bond formation between protein cysteine residues (S-S). All of these can modify the structure and activity of proteins [11]. These protein modifications are reversible through the activities of thioredoxin (TRX) and glutaredoxin (GRX). However, prolonged and/or higher levels of H_2_O_2_ exposure can cause thiol sulfinylation (SO_2_H) and sulfonylation (SO_3_H), which irreversibly damage the protein.

### 2.1. ROS Control Mechanisms: Striking a Fine Balance

A battery of mechanisms exists in cells to keep ROS levels within the non-toxic range, and these mechanisms are exquisitely well regulated to simultaneously allow low levels of ROS to act as signaling molecules. This is particularly relevant in tumor cells, as described below.

The well-recognized enzymes directly neutralizing ROS include the SODs (converts superoxide to H_2_O_2_); catalases (converts H_2_O_2_ to H_2_O and O_2_); and glutathione peroxidases (converts H_2_O_2_ to 2 molecules of H_2_O). Conversely, the hydroxyl radical cannot be detoxified enzymatically, contributing to its detrimental effects in cells.

GSH is often referred to as the most important non-enzymatic antioxidant in cells, as it is essential in various antioxidant processes. GSH is found throughout cellular compartments in millimolar concentrations (1–10 mM). GSH contributes to the maintenance of ‘healthy’ levels of ROS in several ways (Figure 2). First, GSH is involved in the regeneration of enzymatic and non-enzymatic antioxidants; this includes the regeneration of the glutathione peroxidases (GPXs) that detoxify lipid hydroperoxides and H_2_O_2_. GSH is directly used in glutathione-S-transferase (GST) reactions in the detoxification of xenobiotics or products of oxidative stress [12]. The reversible glutathionylation/deglutathionylation reactions catalyzed by the glutaredoxin (GRX) enzymes also directly use GSH [13,14]. In addition, GSH supports the non-enzymatic regeneration of alpha-tocopherol, which prevents lipid peroxidation in cellular membranes. Finally, GSH can directly neutralize superoxide anion radical [15]. As GSH is used to support all of the latter processes, it is regenerated by reactions including the reaction catalyzed by glutathione reductase (GR), which requires NADPH as a coenzyme. NADPH is produced through the oxidative pentose phosphate pathway and through the activities of isocitrate dehydrogenase-1 (IDH1; cytoplasm), isocitrate dehydrogenase-2 (IDH2; mitochondria), malic enzyme-1 (ME1; cytoplasm), malic enzyme-3 (ME3; mitochondria), aldehyde dehydrogenase-1-L1 (ALDH1L1; cytoplasm), aldehyde dehydrogenase-1-L2 (ALDH1L2; mitochondria) and methylenetetrahydrofolate dehydrogenase-1-like (MTHFD1L; mitochondria) [16]. The mitochondrial inner membrane nicotinamide nucleotide transhydrogenase (NNT) also produces NADPH and this serves to couple reductive stress-induced H_2_O_2_ production to NNT-associated NADPH and glutathione redox mechanisms [17]. Beyond acting as a cofactor for GR, NADPH is also essential for the activity of thioredoxin (TRX) and peroxiredoxin (PRX) [18,19]. Altogether this series of reactions enable the SODs to convert superoxide to H_2_O_2_ and the peroxiredoxins (PRXs) and glutathione peroxidases (GPXs) to convert H_2_O_2_ to H_2_O.

The widely dispersed cellular localizations of the various isoforms of these enzymes normally protect cells very well from oxidative damage. Relevant to the central role of GSH, glutathione peroxidase 4 (GPX4) plays an additional protective role in catalyzing the reduction of organic hydroperoxides, and lipid peroxides in membranes. Finally, under oxidized redox potential conditions, the oxidized form of GSH can reversibly bind to protein thiol residues to protect them from irreversible oxidative damage, and/or modify their activity. Some of our previous research demonstrated that the glutathionylation of the mitochondrial uncoupling proteins UCP2 and UCP3 contributed to the control of mitochondrial ROS levels [9,20,21,22]. Mitochondrial protein glutathionylation has been the subject of a number of recent reviews [23,24,25].

### 2.2. RNS Control Mechanisms

One important example of reactive nitrogen species (RNS), is NO, a reactive gas produced by a family of enzymes known as nitric oxide synthases (NOS), namely endothelial NOS, neuronal NOS and inducible/macrophage NOS. Due to the high concentration of GSH in the cells, NO rapidly reacts with the cysteine thiol group of GSH to form S-nitrosoglutatione (GSNO). GSNO represents a reservoir of NO bioactivity and a delivery system of NO to maintain cellular homeostasis [26]. As a scavenger of toxic free radicals, GSNO is 100-fold more potent in neutralizing ROS than GSH and is further metabolized by an evolutionary conserved enzyme S-nitrosoglutathione reductase (GSNOR) at the expense of NADH to generate GSSG and ammonia (Figure 3). GSNOR is a member of the class III aldehyde dehydrogenase (ADH5), also known as glutathione-dependent formaldehyde dehydrogenase (GSH-FDH) and exhibits its highest catalytic activity for GSNO reduction. GSNO has been shown to be an inhibitor of glutathione reductase and a substrate for the thioredoxin system [27]. Thioredoxin can mediate denitrosylation of target proteins by removing the NO moiety, in turn, converting thioredoxin to the disulfide form in an NADPH-dependent reaction (Figure 3).

NO can also react with sulfhydryl moieties of cysteines in proteins to form S-nitrosoproteins or S-nitrosothiols in a process known as S-nitrosation, which results in a change in the activity and function of proteins. Similar to other redox signaling mechanisms such as phosphorylation, acetylation and ubiquitylation, S-nitrosation has emerged as an important redox-based post-translational modification of proteins [26,28]. A recent investigation has shown that more than 3000 proteins are constitutively S-nitrosated, which are involved in cell signaling and regulation of tissue homeostasis [26,28]. For instance, protein nitrosation has been implicated in the regulation of a variety of protein functions and cellular activities including, apoptosis, iron homeostasis, metabolism, redox balance and gene transcription [27,28]. Although in the last two decades the role of GSNOR in regulating protein nitrosation and its impacts on cellular homeostasis and pathophysiological processes has been established, the precise mechanisms of selective denitrosation still remains to be uncovered.

## 3. Glutathione, Redox and ROS in Healthy and Tumor Cells

Redox and ROS are also of particular importance in tumor cell signaling pathways associated with cancer progression through their roles in tumorigenesis, tumor cell migration, and tumor cell survival [29,30,31]. Compared to healthy non-malignant cells, tumor cells have aberrant ROS homeostatic characteristics. Most cancerous cells have a higher ROS set point than associated non-cancerous cells. In many types of tumor cells these higher levels support their growth, proliferation, metastasis and survival in various microenvironments or conditions. Low to medium levels of ROS can lead to genetic instability and tumor suppression (see recent reviews of [32,33]).

Tumor hypoxia is well known to drive higher ROS production [34]. The higher levels of ROS can also be caused by oncogenes such as constitutively active isoforms of *RAS*, and oncogenic K-RAS have been found to be necessary for precancerous lesions in the pancreas and for lung adenocarcinoma formation [35,36].

In many cancers associated with mutations in genes encoding electron transport chain proteins there is increased ROS production that is associated with resistance to apoptosis. For instance, impaired Complex I function associated with *ND6* mutations leads to high levels of ROS generation and a highly metastatic phenotype that can be prevented by ROS scavengers [37]. As well, an increased tumorigenic potential is associated with a heteroplasmic *ND5* mutation that results in apoptosis resistance and increased activity of the PI3K/Akt pathway [38]. There is evidence that the full inhibition of Complex I that prevents increased ROS generation impairs hypoxic adaptation in tumor cells by rewiring mitochondrial metabolism, and the antitumorigenic effect of ROS signaling is lost [39]. Thus, the balance between mitochondrial electron transport chain activity and associated ROS production is crucial.

The increased ROS levels in some tumor cells are also commonly associated with decreased expression of cellular antioxidant systems. For example, the loss of tumor suppressors such as p53 can contribute to tumorigenesis through the elevated ROS levels occurring because of the cellular antioxidant systems’ loss. p53 increases the expression of SOD2 and GPX1 as well as NADPH production [39,40,41].

However, the increased production of ROS by many tumor cells occurs concomitantly with increased expression and activity of antioxidant systems. This may be related to the findings that high levels of ROS can mitigate metastasis and tumorigenesis, for example in cancer cells detaching from the extracellular matrix [42,43]. Indeed the upregulation of NRF2 (nuclear factor, erythroid-derived 2-like 2 factor) has been observed in a wide array of tumors including breast, ovarian, prostate, skin, lung and pancreas [32,44]. NRF2 is a master regulator of antioxidant response as a transcription factor regulating expression of several enzymes responsible for glutathione synthesis, including glutamate-cysteine ligase modifier subunit (GCLM) and the glutamate-cysteine catalytic subunit (GCLC), as well as enzymes related to GSH utilization, such as glutathione reductase, glutathione peroxidase and glutathione S transferase (GST) [45,46].

Elevated levels of GSH have been observed in many cancers including breast, ovarian, lung, as well as head and neck cancers [47]. Through its many physiological functions (Figure 4) glutathione has a major role in tumors with a number of publications showing that increased cellular levels of glutathione are necessary for tumor initiation and proliferation [48,49].

The observed increase in GSH levels, is due not only to the higher levels of ROS production in most tumor cells, but it is again related to the fact that some of the classical tumor promoters also activate GSH synthesis and turnover mechanisms (e.g., NRF2) [51]. In tumors associated with mutated KEAP1 there is increased NRF2 and flux of glutamine to glutamate for GSH synthesis [51,52]. The expression of the cystine/glutamate antiporter (Xc-; SLC7A11), which is needed to support the flux of cysteine into GSH synthesis has markedly increased expression in human tumors [53]. Moreover, NADPH production by the oxidative branch of the PPP is increased in many types of cancer cells [32,53]. Recent research revealed that estrogen-related receptors (ERRs) act as novel redox sensors and effectors of a ROS defense program in breast cancer. Specifically, it was demonstrated that ERRs control glutamine utilization and glutathione antioxidant production [54]. In addition, the expression of GCLM and SLC7A11 genes are regulated by HIF-1, with HIF-1 thereby controlling glutathione biosynthesis in hypoxic environments [55].

### 3.1. Glutathione Synthesis: Normal versus Tumor Cells

GSH biosynthesis is a well-conserved pathway in which the three precursor amino acids, cysteine, glutamate, and glycine, are combined to form the tripeptide GSH. In animals, this process occurs exclusively in the cytosol, where the two enzymes glutamate cysteine ligase (GCL) and glutathione synthetase (GS) are localized. GCL catalyzes the first, and rate-controlling, step of the pathway that creates a linkage between glutamate and cysteine via the γ-carboxyl residue of the latter to produce γ-glutamylcysteine [56]. ATP hydrolysis and Mg^2+^ as a cofactor are required to facilitate this reaction. Furthermore, the GCL enzyme is composed of a catalytic and modifier subunit (GCLC and GCLM, respectively); GCLM is subject to feedback inhibition by GSH. The second and final step of the pathway produces GSH via the addition of glycine to the γ-glutamylcysteine intermediate. [57]. Coupled to this is the degradation of GSH to its constituent amino acids, where a γ-glutamyl cyclotransferase enzyme hydrolyzes GSH to produce the intermediates cysteinyl glycine and 5-oxoproline. Cysteinyl glycine is subsequently broken down into cysteine and glycine through cleavage by its respective peptidase, and a 5-oxoprolinase enzyme produces glutamate through ATP hydrolysis [58]. These pathways form the glutathione (also known as γ-glutamyl) cycle, as shown in Figure 5.

It is well-known that shifts in GSH metabolism often accompany tumor development, potentially as a way for the cancer cells to mitigate the impact of increased oxidative stress that results from their extraordinary metabolic rates [59]. For example, despite differences in methodology, many types of cancers such as ovarian, breast, and lung, consistently display elevated levels of GSH [47]. Surprisingly however, tumors in the brain and liver seem to not need this added antioxidant defense, as suggested by their markedly low levels of GSH [47]. Some examples of the transformed GSH metabolism in cancers can be seen in renal cell carcinoma, where there are increases in both expression and activity of GCLC and GCLM, causing elevated levels of GSH [60]. This observation is not surprising considering the loss of function of pVHL in a number of renal cancers leading to stabilization of HIF-α and subsequent “pseudohypoxic” possibly resulting in an increased activity of GCLC and GCLM. We have in our earlier work shown the relationship between VHL loss of function and cell metabolism and glutathione [55]. Furthermore, within squamous cell carcinoma there is a dramatic increase in the expression of many enzymes of the glutathione metabolic pathway related to its antioxidant function and synthesis [61]. These changes in GSH content seen in cancerous cells have been suggested as essential for the tumor’s development, and drug resistance [54,58].

Metabolic heterogeneity, resulting from factors both intrinsic and extrinsic to the cancer cell, leads to tumor subtype specific metabolic properties and vulnerabilities [62]. Diversity in metabolic phenotypes is observed even in tumor cells with similar genetic background or tissue of origin [63,64]. Large analysis presented in [65] shows that across measurement of 928 different cancer cell lines from 20 major cancer types, glutathione ratio (GSH/GSSG) as well as NADP+ are most strongly associated with KEAP1 mutation. GSH and GSSG values and the ratio GSH/GSSG provided by [65] are shown in Figure 6.

### 3.2. Transport of GSH

The range of functions of GSH illustrate the great demand for this molecule not only throughout the cell and its organelles, but also in the surrounding extracellular environment. Due to the properties of GSH (namely its size and net negative charge), GSH transport generally requires some form of carrier to cross the plasma membrane, or the membranes of organelles to reach its many targets. Additionally, GSH is not only the most prominent non-protein thiol in cells, it is also one of the most abundant metabolites, which further illustrates the demand for transport [66]. Unfortunately, despite the vast amounts of GSH needed to be transported into and out of organelles and cells, knowledge of transporter proteins in humans is still not complete [67].

Some of the earliest investigations into the membrane transport of GSH came with the transporter, multidrug resistance protein 1 (MRP1). The MRP or ABCC family of proteins are a group of transporters found on the plasma membranes of many cells. Early studies found that MRP1 could provide tumor cells with multidrug resistance (MDR) by exporting xenobiotics, and that this function was associated with GSH [68]. The details of the mechanism by which this occurs is poorly understood at a molecular level, however as mentioned above, it is largely facilitated through export of GSH conjugates of the targets [69]. Furthermore, recently 8 other members of this family (MRP2-9) have been discovered, each with evidence to support GSH conjugates, and other GSH species, as substrates [70].

Several organelles have been demonstrated to require their own GSH pools, including the nucleus, endoplasmic reticulum, and mitochondria. These organelles generally have similar concentrations of GSH as in the cytosol, although their redox ratio (reduced GSH to oxidized GSSG) may vary [71]. Mitochondria, for example, contain 10–15% of the total cellular GSH content [72], where the OXPHOS-mediated generation of ROS results in a large demand for GSH. The influx of GSH into mitochondria requires passing through both the outer and inner mitochondrial membranes (OMM and IMM, respectively), however movement across the OMM is kept in equilibrium with the cytosol for small molecules such as GSH via free movement of small molecules (<5 kDa) through porin proteins in the OMM [73]. Transport across the IMM however requires specific proteins. Previously, two transporters had been identified to carry out this function, the dicarboxylate and the 2-oxoglutarate carriers (SLC25A10 and SLC25A11, respectively); however, this has recently been put under doubt [74].

Due to its role in DNA repair and protection from oxidative damage, the nucleus seems like an obvious target for GSH localization. Indeed, the antioxidant plays an important role within the nucleus and in cell proliferation [75,76]. Interestingly, the nuclear GSH pool appears to be dynamic and independent of cytosolic GSH, undergoing major shifts in relative concentration throughout the cell cycle which reaches upwards of four times the concentration found in the cytosol [76]. The independence and differences of the nuclear GSH pool, coupled with the tight regulation of transport across the nuclear envelope, suggests specific GSH carrier proteins, much like in mitochondria. Experiments have reinforced this idea, disproving nuclear synthesis of GSH or passive movement across membranes [77]. Unfortunately, there are yet no identified nuclear membrane transporters of GSH. There has been some support for the idea that Bcl-2 along with the 2-oxoglutarate carrier function as GSH transporters, based on correlations between Bcl-2 expression and nuclear GSH levels in cancer, however, this idea has been abandoned since the 2-oxoglutarate carrier has been refuted as a GSH transporter [76,78].

Unlike in other organelles, investigations into the mechanisms behind endoplasmic reticulum (ER) GSH transport have been more successful in identifying the proteins responsible. The channel protein Sec61 has been demonstrated to import and export ER GSH, and transport is modulated by Bip and Ero1 [79]. Interestingly, a recent study has shown that xylitol, a compound found in the fungus *Cordyceps militaris*, may be a promising anticancer therapy via regulation of GSH [80]. In the proposed mechanism, xylitol induces Bip and CHAC1, resulting in decreased cellular GSH levels, ER stress, and ultimately apoptotic cell death. Furthermore, xylitol reduced chemotherapy drug resistance when coupled with other therapeutics. However, further research is required to determine whether modulation of ER transport of GSH specifically is responsible for these therapeutic effects.

## 4. Glutathione as an Antioxidant

Another important GSH pathway is the antioxidant pathway via the mitochondrial enzyme GPX4, which is responsible for the elimination of damaging lipid peroxides, and other ROS species. This pathway is of particular relevance to disease as oxidative stress is a hallmark of many cancers, neurodegenerative disease, and more; one result of this stress being lipid oxidation [81]. Moreover, high levels of certain metals, like Fe, Ni, and Cu, can initiate oxidative stress by reacting with naturally occurring hydrogen peroxide to produce hydroxyl radicals [82]. Along with protecting against their downstream effects via GPX4, GSH is capable of directly conjugating many of these metals/metal complexes to neutralize them [82]. Combined, GSH provides an excellent tool for the cell protection against these stresses.

Furthermore, GPX4 is an essential enzyme in the recently discovered ferroptosis cell death pathway. A major result of this is the depletion of GSH, and disruption of the GPX4 antioxidant pathway, leading to massive lipid peroxidation and cell death [83]. GPX4 is a major regulator of this form of programmed cell death, which is further exacerbated in cancers where recent studies have shown GPX4 as essential in some tumors [55,61]. Much like other cell death pathways, ferroptosis is often deregulated in tumors and thus may be a viable target for cancer therapy, with for example small molecule drug erastin targeting cysteine import system (described below). Ferroptosis is affected by several metabolic pathways including lipid oxidation metabolism, glutathione metabolic pathway and iron metabolic pathway through regulation of oxidation of phosphatidylethanolamines (PEs) containing arachidonic acid (AA) or adrenic acid (AdA) to phospholipid hydroperoxides (PE-AA/AdA-OOH). Accumulation and insufficient removal of these lipid peroxides (LPO) as well as intracellular iron are fundamental for ferroptosis. The cystine availability and glutathione (GSH) biosynthesis regulating GPX4 function are main regulators of redox homeostasis there-by protecting cells from LPO accumulation. Ferroptosis is of increasing relevance for cancer therapy as well as understanding of the development of tumor and other degenerative diseases such as Alzheimer’s and Parkinson’s disease. Several recent reviews have described in some detail current understanding of ferroptosis and role of GSH in this process [84,85].

### Glutathione S-Transferase

As mentioned above, cellular damage may arise from small molecules including metals such as iron and copper. Protecting against these, and other, potentially toxic species is another challenge faced by the cell. A superfamily of enzymes known as glutathione transferases (GSTs) utilize GSH to combat a wide range of xenobiotics and other harmful compounds [86]. These enzymes are broadly separated into three classes based on their cellular location: cytosolic GSTs, mitochondrial GST, and microsomal GSTs (also known as MAPEG GSTs) [87]. Due to the number of members in this superfamily, there is a wide range of functions associated with these enzymes including: glutathione peroxidase activity, prostaglandin synthesis, and roles in thiolysis and isomerization reactions [86]. Additionally, GSTs perform several non-enzymatic functions, such as signaling in the MAPK pathway to modulate apoptosis [88].

Most relevant of these are the previously mentioned xenobiotic reactions, in which GSTs function as Phase II detoxification enzymes to eliminate a variety of hydrophobic and electrophilic compounds via GSH [65,67]. The targets of GST facilitated detoxification range from the apoptosis signaling molecule 4-hydroxynonenal, to various insecticides and anti-cancer drugs like cisplatin [67,89]. The mechanism by which this detoxification occurs is through the addition of GSH to the harmful compounds via a reaction at an electrophilic residue and the sulfhydryl residue of GSH [90]. The resulting GSH conjugated compounds are unable to effectively damage the sensitive cellular components and can then be efficiently exported out of the cell.

Within cancer, GSTs play a significant role in both tumorigenesis, as well as in established cancer cells. Interestingly, in tumorigenesis these enzymes protect from cancer development but once this fails GSTs are essential for ensuring the survival of tumor cells [91]. For example, mice with silenced GSTp1 (also known as GST pi 1), a cytosolic GST, have a six-fold increase in colon cancer development compared to their wild-type counterparts [92]. Furthermore, polymorphisms of the various GSTs have been associated with susceptibility to many diseases including several cancers [93,94]. This preventative role of GSTs in tumorigenesis is due to the mitigation of DNA damage caused by xenobiotics, carcinogens, and oxidation that could otherwise cause oncogenic mutations if not for the detoxification and peroxidase activities of this family of enzymes. Along with this, the signaling role in the MAPK pathway adds to the sensitivity of the cell to excessive damage or mutation, leading to apoptosis [88].

GSTs have been linked to MDR in tumors, a common problem in treatment of many cancers. Members of the super family have long been known to be overexpressed in many types of cancers [95,96,97]. As previously mentioned, the cytosolic GSTp is a prominent member of the cancer-associated GSTs [95,96,97]. Interestingly, overexpression of this enzyme is not seen in breast cancer cells, yet MDR is a common feature of the disease [98]. This led researchers to look at the tumor microenvironment and found significant increases in GSTp [95,96,97]. Recently, microenvironment GST has been suggested as important to metastasis and apoptosis in ovarian and bladder cancer [99,100]. Polymorphisms of GSTs have been identified which affect the expression or activity of the enzymes and have been associated with many diseases; some of these have since been proposed as biomarkers of cancer risk and prognosis [78,79,80].

## 5. Glutathione, Inflammation, and Cancer

Chronic infection, injury and inflammation have long been recognized as risk factors for the development and progression of various human cancers [101]. There is ample evidence to indicate that factors present in the tumor microenvironment (TME), such as products of inflammatory cells reactive nitrogen and oxygen species may cause DNA damage to the neighboring cells, thereby contributing to tumor progression [102]. A detailed review of many roles of the inflammation-driven changes in tumorigenesis has been recently provided [102]. It has been established that GSH deficiency or a change in GSH/GSSG ratio increases the vulnerability of cells to oxidative stress, inflammation, and tumor progression. However, elevated GSH levels increase antioxidant capacity and resistance to oxidative stress as is evident in many tumors. It has been demonstrated that exogenously added GSH inhibits the inflammatory response through regulation of ROS while endogenous GSH has been recently shown to have a role in fine-tuning the innate immune response to infection thereby regulating inflammation [103]. GSH therefore has a dual role in the inflammatory response as an antioxidant ROS scavenger in the oxidative stress as well as signaling molecule that regulates protein function through thiol–disulfide exchange reactions such as protein glutathionylation with many examples of regulation of oncogenes (e.g., p53, HIF-1, c-jun) presented in the database developed by Chen et al., [104] as well as in a recent review [105]. Here we will focus on the role of different antioxidant enzymes, specifically peroxiredoxin and glutathione peroxidases as well as the impact of NO-mediated post-translational modification of oncoproteins on tumor progression.

### 5.1. Role of Peroxiredoxin (PRDX)

PRDX is a family of ubiquitously expressed proteins that catalyzes reduction of H_2_O_2_, organic hydroperoxides and peroxynitrite [106]. In human cells there are six different isoforms of PRDX with specific localization in the cytosol, mitochondria, nuclei, peroxisomes, endoplasmic reticulum, as well as secretome. In all isoforms enzymatic function depends on the conserved peroxidatic cysteine residue contained within an active-site conserved across all isoforms as well as cysteine in the carboxy-terminal region functional in some of the PRDX isoforms. Antioxidant function of PRDX is typically performed through reaction between the main cysteine residue and the residue on the second subunit of the dimer. GSH together with thioredoxin can act as a physiological reductant in this process. Specific roles of different PRDX isoforms in tumor-promoting, tumor-suppression as well as tumor treatment resistance have been reviewed previously for different cancers [106] with a number of specific applications to different cancer types including for example breast [107], lung [108] or pancreatic cancers [109]. In addition to the direct role in redox processes, PRDX in relation to glutathione has a major link to inflammation leading to cancer.

Inflammatory stimuli cause release of oxidized peroxiredoxin-2 (PRDX2), a ubiquitous redox-active intracellular enzyme that extracellularly acts as a redox-dependent inflammatory mediator. The glutathionylation of GSH to PRDX2 cysteine residues has been shown to be central to the regulation of immunity. Additionally, several studies show presence of PRDXs in body fluids of cancer patients, suggesting that PRDXs and its glutathionylation may be a link between inflammation and cancer [110]. Additionally, ROS-induced cysteine glutathionylation and its modulation have been shown to be the key regulators of IL-1beta bioactivity by preventing the irreversible ROS-elicited deactivation [111]. IL-1beta is increasingly understood as a major regulator of tumor promoting inflammation through its activity on different components of tumor microenvironment including recruitment of tumor infiltrating myeloid cells, angiogenesis, and suppression of anti-tumor immune response [112,113].

### 5.2. Role of Glutathione Peroxidases (GPXs)

Tumor cells have increased ROS levels and thus upregulate the antioxidant response, including glutathione peroxidases (GPXs). GPXs are members of a multi-enzyme family that catalyze the reduction of H_2_O_2_, organic hydroperoxides and/or lipid hydroperoxides to generate their corresponding lipid alcohols and water at the expense of oxidation of two molecules of GSH [114]. There are eight isozymes of GPXs (GPX 1–8) that have been identified in humans in different tissues, among these five are selenoproteins (GPXs 1–4 and GPX6) that play an important role in defense against oxidative stress.

Cellular models and transgenic mice deficient in individual *GPX* genes have been utilized to study the potential role of individual GPXs in malignancy. Several in-depth reviews have been published on the molecular mechanisms of GPXs regulation in tumors and their function in different stages of tumor progression [115,116,117]. GPX1 and 2 are highly expressed in the gastrointestinal epithelium and have been implicated in the development and promotion of colorectal cancer [8]. For instance, mice deficient in both *GPX1* and *GPX2* genes, spontaneously developed ileocolitis at a young age. The ileum of these mice lost GSH-dependent activity to reduce peroxides and had increased expression of *NOX1* mRNA and elevated levels of TNF, suggesting that NOX1 is the major source of ROS and further supporting the anti-inflammatory function of GPX1 and GPX2 [118]. Using GPX1 and GPX2-deficient mice, a recent study has shown that loss of GPX2 resulted in a robust NFkB activation and release of IL-1beta as compared to GPX1 mice. However, loss of GPX1 induced the expression of cyclooxygenase-2 and release of prostaglandins [119]. Taken together, these studies demonstrate that GPX1 and 2 have important roles in regulating the synthesis and release of proinflammatory mediators.

GPX3 is the only extracellular GPX that detoxifies lipid hydroperoxides at the expense of GSH. GPX3 mRNA is expressed in the kidney, pancreas, lung, breast, brain and gastrointestinal tract, but the majority of GPX3 is expressed in the plasma which is kidney-derived. It has been reported that GPX3 can act both as a tumor suppressor and a pro-survival protein. Lower GPX3 levels in the plasma and tumor tissue of non-small-cell lung cancer, glioblastoma, hepatocellular carcinoma and colorectal carcinoma have been linked to selenium deficiency and increased lipid peroxidation, supporting the notion that loss of GPX3 contributes to oxidative stress [120]. GPX3 loss in the plasma of cancer patients is often associated with poor patient outcome. The exact mechanism of GPX3 loss is not known but might be associated with enhanced utilization of selenium and glutathione by tumor cells. Systemic GPX3 loss in mice was shown to promote tumor initiation and accelerate inflammatory colonic tumorigenesis [121]. It is noteworthy that GPX3 loss was not associated with increased ROS levels or DNA damage, however exposure to carcinogens resulted in an increase in M2 macrophages, elevated proinflammatory mediators and increased DNA damage in tumor cells.

There are also studies demonstrating that GPX3 expression is increased in certain cancers, including clear cell adenocarcinoma, colorectal carcinoma, ovarian cancer and leukemia [120]. Hence the role of GPX3 in tumor progression is controversial and further mechanistic studies are needed to achieve specific targeting.

GPX4 is also an important enzyme that protects membranes against lipid peroxidation [122]. It has emerged as a central regulator of a recently discovered form of cell death known as ferroptosis which is induced by cytosolic and lipid ROS [83]. Depletion of *GPX4* in cells and mice resulted in increased lipid-based oxidative stress and induction of 12/15-lipoxygenase, lipid peroxidation and apoptosis-inducing factor-mediated cell death [123]. GPX4 dampens inflammation by decreasing lipid-based oxidative stress produced by the arachidonic acid metabolites and NFkB pathway [124]. While the role of GPXs 1,2,3 and 4 in tumorigenesis is established, the knowledge on GPXs 6–8 is limited and should be systematically investigated.

### 5.3. Role of S-Nitrosation

NO is produced enzymatically by nitric oxide synthases (NOS) not only by tumor cells but also by infiltrating inflammatory immune cells and stromal cells [125,126]. As mentioned previously, NO carries out its biological function mainly through protein S-nitrosation, a redox modification in which NO is covalently attached to the thiol group in cysteine residues of almost all functional classes of proteins. Increasing evidence suggests that dysregulated S-nitrosation, which could result from alterations in the expression or activity of NOS and denitrosylases, including GSNO reductase has emerged as an important mechanism promoting tumor progression. Several human tumors, including breast, ovarian, pancreatic, liver, lung, glioblastoma, prostate, melanoma and colorectal cancer often express increased eNOS, nNOS and iNOS, however, tumor-infiltrating immune cells mainly express iNOS [127]. A large number of oncogenic proteins, including epidermal growth factor (EGFR), mitogen-activated protein kinase phosphatase-1, tumor necrosis factor receptor associated protein 1, phosphatase and tensin homolog (PTEN), p53, ras and src tyrosine kinase have been shown to be directly activated by S-nitrosation, which might have diverse functions in different stages of tumor progression [128]. S-nitrosation of EGFR and src resulted in activation of oncogenic signal transduction pathways, including c-Myc, Akt, and β-catenin in human basal-like breast cancer [129]. These signaling pathways are abnormally activated in various types of breast cancers and may offer survival advantage and promote invasion and metastasis. It was found that aberrant S-nitrosylation increased survival of HER^2+^ breast tumors as well as resistance to trastuzumab [130]. More recently, Ehrenfeld et al., demonstrated that S-nitrosation of endothelial cell proteins might be an important mediator of tumor angiogenesis, invasion and metastasis [131]. Therefore, novel therapeutic development is focused on promoting or inhibiting S-nitrosation or restoring a balance between S-nitrosation and denitrosation in tumor cells, thus specifically targeting tumor progression.

## 6. Glutathione and Tumor Microenvironment

Solid tumors can be considered as complex organs composed of not only tumor cells but also a diverse population of non-tumor cells that form the physical components of the tumor microenvironment (TME). The non-tumor cells include stromal fibroblasts, also known as cancer-associated fibroblasts (CAFs), tumor vasculature consisting of endothelial cells and pericytes as well as infiltrating immune cells consisting of lymphocytes, neutrophils, macrophages, and mast cells. The physiological characteristics of the TME are defined by its pH and oxygen tension as well as presence of nutrients, cytokines, chemokines, growth factors, matrix remodeling enzymes, and metabolites [132]. Tumors have evolved to create unique networks in which tumor cells communicate not only with each other but also with stromal and immune cells via cell-cell contact and secreted factors to evade immune responses. Among the secreted factors, extracellular vesicles (EVs) and mainly exosomes have emerged as novel communication vehicles that have been detected in the TME and are increasingly understood as key players in tumor growth and spread [133,134]. It has increasingly become evident that EVs, including exosomes could participate at several stages of tumor progression, from initiation, invasion to metastasis.

Exosomes are nanometer-sized (~40–150 nm) membrane-enclosed vesicles that are derived from the endosomes through inward budding to generate multivesicular bodies (MVB). The MVB fuse with the plasma membrane to release EVs in the TME. It has now become clear that essentially all cell types constitutively secrete exosomes into the extracellular space to mediate intercellular communication [135]. Within the TME, exosomes convey bioactive messages in the form of nucleic acids (miRNA, mRNA, non-coding RNAs and mitochondrial DNA), proteins (cytokines, receptors and receptor ligands), metabolites and lipids, reflecting the metabolic state of the cells from which they are derived [136,137]. Both tumor and stromal cells release exosomes that carry ‘specific addresses’ and are selectively internalized by recipient cells to modulate the TME locally or travel through the circulation to other organs [138]. Tumor cells produce higher numbers of exosomes and exploit them to promote tumor growth by reprogramming the TME that favors tumor proliferation, invasion and metastasis [138,139]. Another remarkable property of tumor-derived exosomes is their ability to facilitate the formation of a pre-metastatic niche, microenvironment supporting metastasis. Peinado et al., has elegantly demonstrated that mouse melanoma-derived exosomes were able to educate and reprogram bone marrow cells to a melanoma-promoting environment. More importantly, exosome-mediated changes resulted in dysregulated signaling that interfered with hematopoietic cell development, differentiation and function [140]. In another landmark study these authors showed that tumor-derived exosomes were internalized by organ-specific cells and was associated with the formation of a pre-metastatic niche, dictating organotropic metastasis [141].

Over the past decade, research efforts in tumor biology have highlighted the pivotal role of exosomal oncogenic proteins and RNAs to modulate the TME and support tumor progression [142,143,144]. The role of metabolic reprogramming in tumor cells has now gained attention with the realization that oncogenic mutations in isocitrate dehydrogenase (IDH) 1 and 2 have important consequences in establishing metabolic phenotypes in glioma cells [145]. Similarly, mutations in other oncogenic genes such as p53 and c-myc have been shown to trigger glycolysis and facilitate lactate production. Interestingly, oncogenic proteins have been found in tumor-cell derived EVs which could remodel recipient tumor and stromal cells in the TME to favor tumor progression [146].

Recent studies have revealed that CAF-derived exosomes in the TME can facilitate communication between tumor and stromal cells and contribute significantly to tumor progression [147,148]. There are only a few studies available on the metabolite composition of exosomes and this area remains largely understudied. CAF-derived exosomes have been shown to contain amino acids, lipids and TCA intermediates that inhibit mitochondrial oxidative phosphorylation, thereby increasing glutamine-dependent reductive carboxylation and promote glycolysis in tumor cells [149]. In addition, CAF-derived exosomes obtained from prostate cancer patients and cultured under in vitro conditions were able to reprogram the metabolic machinery of cancer cells [149]. More recently, EVs derived from glioblastoma (GBM) cells were shown to facilitate transformation of reactive astrocytes and promote tumor growth [150]. GBM-EVs were internalized by reactive astrocytes and appeared to reprogram astrocyte metabolism by delivering mRNAs for glycolytic enzymes (enolase 1, hexokinase-1, and hexokinase-2) and oxidative phosphorylation (cytochrome c oxidase) to activate the biosynthetic processes needed for tumor growth [150]. Taken together, these studies provide support for the role of EVs in metabolic reprogramming and alteration of the TME to promote tumor growth (Figure 7).

Tumor hypoxia and acidosis have emerged as important physiological factors that can affect the redox status, including elevated concentrations of GSH in the TME [150,151,152,153]. Indeed, tumor cells in the hypoxic microenvironment have elevated demands for glycolysis, in turn, leading to increased lactic acid and acidosis. A low pH of 6.5 in tumor cells has been associated with an increased exosome release and entry into metastatic human melanoma cells, through a lipid-dependent fusion [154]. Experimental evidence from human tumor cell lines derived from either colon, breast, melanoma, osteosarcoma and prostate cancers showed increased exosomal release as the pH was changed from 7.4 to 6.5, supporting the notion that acidic TME is associated with increased exosome production and represents a key phenotype of all human cancers [155]. Using three different glioblastoma (GBM) cell lines, namely LN18, A172 and U118, we have recently shown a clear difference between the metabolic profiles of GBM cells, exosomes and their media [156]. In our analysis, GSH levels were higher in A172 and LN18 cells than in their respective exosomes, while GSSG was concentrated in exosomes from all three GBM cells, possibly to remove GSSG from cells. Multiple lines of evidence suggests that tumor cells have evolved to survive within the hypoxic and acidic environment, in part, by upregulating GSH-dependent antioxidant enzymes, including glutathione peroxidases and glutathione reductase and GSH content [157]. However, studies on the characterization of the GSH antioxidant system in exosomes are lacking. With the increasing importance of exosomes in the TME, future studies should be directed to insightful understanding of the metabolic and lipidomic composition of the exosomes under hypoxic and acidic microenvironment and their contribution to tumor progression.

## 7. Examples of Glutathione-Related Tumor Therapies

Following cancer onset, glutathione’s antioxidant function can become superfluous in some cancer subtypes due to thioredoxin antioxidant pathway fulfilling the antioxidant requirements making inhibition of only GSH insufficient for tumor therapy [48]. However, GSH plays a major role in chemotherapy resistance as mentioned above, and inhibition of GSH as part of combined or prodrug therapies has been shown as a very effective approach for improving chemotherapies effectiveness.

Chemotherapy treatment of triple negative breast cancer leads to an increase in expression of GCLM and SLC7A11 and an increase in the intracellular GSH concentration. GSH functions as a signaling molecule which activates the breast cancer stem cell phenotype through copper chelation that leads to an inhibition of mitogen-activated protein kinase kinase (MEK) activity, leading to inactivation of MEK1-ERK signaling, FoxO3 dephosphorylation, and nuclear translocation ultimately leading to enrichment of breast cancer stem cells [158]

High ROS production in cancer cells requires high activity of cellular antioxidant systems making cancer cells hypersensitive to exogenous agents that damage their antioxidant capability. Therefore, reduction in either glutathione production or availability provides an avenue for tumor therapy. Depletion of GSH in cells ultimately leads to ferroptosis [159].

GSH acts as a detoxifying agent by direct conjugation to a xenobiotic (X) forming a glutathione-S conjugate as part of the mercapturic acid pathway that leads to the elimination of toxic compounds. In cancer cells this process can be used for the elimination of chemotherapeutic drugs.

GSH synthesis is primarily controlled by the availability of cysteine, with cysteine starvation leading to GSH depletion. Therefore, inhibition of the cysteine cell transporter system xc− (cystine/glutamate antiporter protein SLC7A11) can lead to GSH depletion and cell death through ferroptosis. This approach is explored with small molecule drug erastin (named after eradicator of RAS and ST-expressing cells) identified to efficiently kill human tumor cells without affecting their isogenic normal cell counterparts [160]. Additionally, depletion of glutathione induced by erastin leads to significantly higher cisplatin’s cytotoxicity and reduced tumor resistance to this chemotherapy [161].

Many cancer drugs exhibit high toxicity making target selectivity a major focus for cancer therapy development. High concentration of GSH in many tumors together with its high reactivity leads towards utilization of GSH as an activator of prodrugs, possibly leading to GSH depletion as well as release of toxic chemotherapeutic agents. An example was presented by Grant et al., [162] with romidepsin that is activated by intracellular GSH by cleavage of a disulfide bond. This reaction produces a thiol group that binds to a zinc atom in Zn-dependent histone deacetylase. Romidepsin is used for treatment of cutaneous T-cell lymphoma (CTCL) and other peripheral T-cell lymphomas (PTCLs).

Development of ROS and GSH responsive nanoparticle-drug delivery systems has been proposed as a way to more specifically and safely deliver highly toxic payload to cancer cells [163]. An example of utilization of this approach was presented by Ling et al., [164]. In this example GSH induced disintegration of optimized nanoparticles allowed release of their payload of active platinum, Pt(II) metabolites that subsequently covalently bound to target DNA and induced apoptosis in cancer cells. Of course, with GSH’s ubiquitous presence in cells and possibly high nanoparticle toxicity [165] this interesting technology needs further optimization prior to clinical utilization.

Alternatively, prodrugs activated by other processes specific to cancer cells can be used for the depletion of GSH. Quinone methide (QM) is an intermediate produced by esterases enzymes and is capable of depleting GSH in cells. As esterases are overexpressed in many cancer subtypes utilization of prodrug esters that are “activated” by the function of esterases into QM to perform GSH depletion is very promising with a number of publications presenting different molecular constructs for prodrug. A recent example of this approach Yoo et al. [166] has presented a novel GSH-depleting pro-oxidant—benzoyloxy dibenzyl carbonate (B2C). In cell and animal models tests, B2C leads to a rapid decline of GSH and elevation of oxidative stress ultimately leading to cancer cells death. GSH depletion is driven through esterase catalyzed release of quinone methide intermediates carboxylic ester of B2C.

## 8. Computer Modeling of GSH Metabolism

Selection of GSH related tumor therapies from analysis of only a subset of its functions can result in highly unexpected results. Computer modeling provides a platform for combining, testing, and predicting glutathione’s many roles as well as testing different routes for its regulation through synthesis, transport, or availability of amino acid components. Computer modeling is an integral process of drug discovery from analysis of optimal targets from large omics data, to modeling, docking or AI driven drug design to modeling of cellular responses to therapy. In cancer applications computer modeling can be used to establish a role of glutathione within cancer metabolism and for providing in silico routines for testing effects of drugs and relevance of drug targets. Mechanistic or machine learning modeling of processes as well as modeling of molecular interactions can be used to explain data or provide hypotheses [167]. Depending on the level of kinetic detail for each step and number of reactions included in the analysis models can go from highly rigorous simulation of a small set of reactions to Boolean or flux models of the whole metabolic network.

One of the most detailed kinetic models of GSH metabolism was presented by Reed et al., [168]. This model provides mathematical representation of one-carbon metabolism, the trans-sulfuration pathway, and GSH synthesis, transport, and breakdown and takes into consideration three compartments—blood, cytosol, and mitochondria with kinetic of reactions as well as transport between compartments. Reed’s model does not include all reactions in GSH metabolism and for several reactions fits parameters to experimental data available at the time, making the model context specific. However, with links between 10 metabolites in 13 reactions simulated using detailed kinetic equations, Reed’s model (available at Biomodels: BIOMD0000000268) is an excellent basis for further development of computer simulations of GSH metabolism for different applications. An interesting example of an extension to Reed’s GSH metabolism model has been presented by Kelly et al., [169] with the addition of quinone redox metabolism to the GSH model. With this addition, glutathione’s role in mitigating toxicity from quinone-derivatives is combined with its overall kinetics providing a link between glutathione depletion in different compartments with quinone redox. Doxorubicin, one of the most important chemotherapy drugs that functions by blocking topo-isomerase 2 is a quinone containing compound. Doxorubicin causes significant dose-dependent cardiotoxicity, possibly due to futile redox cycling. Cellular GSH is a major defense against the redox cycling-derived oxidative stress and therefore a detailed model of glutathione-quinone interplay is of great interest to cancer treatment and analysis of side effects [169].

Several kinetic models have been developed in an effort to describe cancer metabolism with some including a number of reactions involving GSH. An interesting example of such a model published by Roy et al., [170] consists of kinetic models of 53 metabolic reactions within glycolysis, glutaminolysis, tricarboxylic acid cycle and the pentose phosphate pathway represented as a set of ordinary differential equations fit to publish enzyme knockdown experimental data for pancreatic cancer. Of course, this more general model of cancer metabolism can be used for analysis of GSH-related processes as well.

One of the major advantages of detailed kinetic models is the possibility to include inhibition or activation regulation of reactions. Analysis of metabolic self-inhibition based on the enzymological data obtained through a century of experimentation results collected in metabolism databases such as BRENDA [171] has shown a major role of metabolites as regulators of enzyme functions [172]. Reduced GSH is one of the most connected inhibitors with an estimated 70 relationships in its inhibitory network in comparison to 90% of enzymes and inhibitors having 20 or less connections. Also, 52 metabolites have been determined in this analysis to act as inhibitors of enzymes in the GSH metabolic pathway. Therefore, complete simulation of metabolic processes would require inclusion of much more detailed consideration of inhibitory reactions. To the best of our knowledge, none of the currently available models of GSH metabolism include extensive consideration of inhibition of enzymes by GSH and other metabolites. At the same time kinetic models, particularly if considering inhibitory relationships, can currently only provide a window on a small section of metabolism and cannot be developed for simulation of the whole network.

Simulation of metabolism of GSH in the context of a complete metabolic network of cancer cells and tissues as well as metabolic interactions between cancer cells and their microenvironment is possible with either Boolean models or much more informative genome scale flux models [173]. Currently the most extensive metabolic network HUMAN1 [174] includes 13,417 reactions, 10,138 metabolites (4164 unique), and 3625 genes. Within this model GSH is involved in 124 reactions in cytosol, 22 in mitochondria, 15 in peroxisomes and 16 in the endoplasmic reticulum, showing its major involvement in a variety of cellular processes. Tissue- or phenotype-specific genome scale models can be derived from the general model with recent examples of cancer genome scale models provided by Pacheco et al., [175]. These authors developed methods for high throughput reconstruction of tissue specific models from genomics data and created over 10000 reconstructions of genome scale models for cancer tissues from RNA-Seq data included in the TCGA (The Cancer Genome Atlas) data set for 13 different types of cancers. In the original publication the models have been used for the analysis of drug targets, but these models allow analysis of the glutathione metabolism in parallel with the other changes in cancer metabolism.

Finally, many tumor tissues include both cancer and healthy cells as well as possibly unique microorganism content within the microenvironment. Therefore, for computational models to provide true in silico laboratories and complete understanding of the role of GSH in this complicated system it will be necessary to simulate molecular single-cell effects in the context of a multicellular system. Current approaches for multi-cellular system modeling for cancer have been recently reviewed [176] and give an interesting basis for future analysis of glutathione’s role in these systems. An example of modeling tumor spheroids by combining multiscale and multicellular models was recently presented by Roy and Finley [177]. The model includes detailed kinetic representation of intracellular metabolism linked within the cluster of tumor cells that simulates tumor spheroid systems. Although the kinetic model of metabolism included in the spheroid model only covers the basic GSH to GSSG oxidation reaction presented approach and system provides an opportunity for further expansion possibly with additional GSH reaction or transfer across cells and organelles.

## 9. Conclusions and Future Directions

GSH exists in organelles, cells, tissues, extracellular vesicles, and body fluids in a reduced form or as a disulphide, in its oxidized form. The high reactivity of GSH makes exact measurement of absolute concentrations challenging, likely leading to many discrepancies in reported concentrations in various biological systems. Introduction of advanced analytical techniques including a variety of mass spectrometry and nuclear magnetic resonance spectroscopy (NMR) approaches and platforms, as well as better methods for prevention of auto-oxidation now leads to more accurate measurement of the changes in GSH/GSSG concentrations. This is particularly important in the development and utilization of tumor therapies where GSH levels can be both an indicator of therapy success and a direct target to therapies. Development of intelligent and targeted tumor therapy that takes advantage of the very high demand of tumors for GSH requires further understanding of the many roles of this important molecule. More accurate measurement of GSH concentrations and its flux across cell organelles and between tumor cells and their microenvironment will allow development of better tools for in silico exploration of therapeutic opportunities and deeper understanding of glutathione’s many roles.

It is also important to recognize that our knowledge of the role of GSH in tumor development, progression and therapy is based primarily on data from tumor or stromal cells grown in isolation under in vitro experimental conditions and caution should be exercised to extrapolate these findings to the complex in vivo TME. Attempts are being made to develop tumor organoid models with the possibility to provide better representation of redox processes in the tumor system and further work in this area is essential. Moreover, exosomes represent a heterogeneous population of vesicles and our inability to distinguish tumor from stromal cell-derived exosomes and their distinct cargoes add another layer of complexity. Therefore, it is necessary to develop a better toolkit and standardize methods of isolation and analysis of specific subset of exosomes from in vivo tumor tissue. Based on the recent studies that exosomes harbor an ‘off-the-shelf’ pool of metabolite cargo which can fuel tumorigenesis, preventing exosomes from providing nourishment to tumor cells might be an effective strategy for cancer treatment.

Improvement in therapeutic targeting will require a better understanding and stronger focus on the TME and inflammation as an integral part of tumor progression with an integrated approach in the analysis of tumor and various stromal cells. Utilizing high-throughput measurement methods applied to the novel, more accurate and appropriate model systems will provide valuable data for the development of GSH metabolism and interaction models both as mechanistic, knowledge-based or machine-learning, data-based models providing platforms for drug target determination and investigation of complete effect of combined therapies. Major developments in the machine learning approaches for drug design including improvement in the drug and target parameterization methods are expected to provide novel, more specific and sensitive treatments and provide avenues for the development of safe multimodal cancer therapies. GSH, with its pleiotropic functions, deserves to have a prominent role in these developments.

## Figures and Tables

**Figure 1 biomolecules-10-01429-f001:**
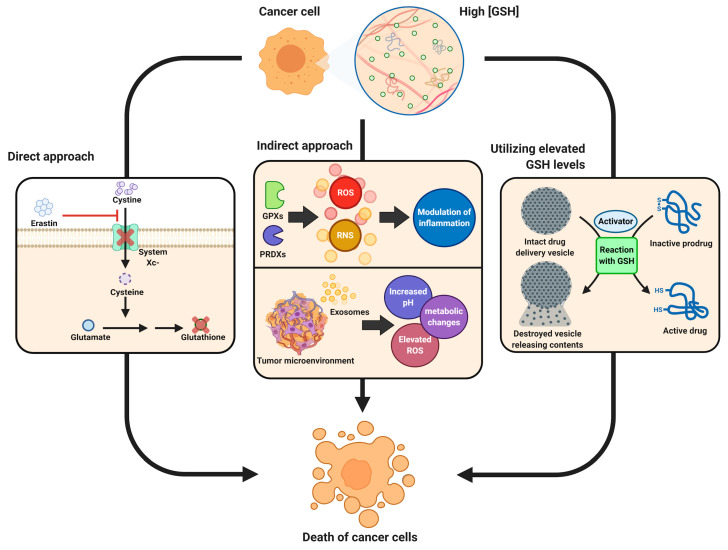
Different approaches for cancer therapy related to glutathione functions. High levels of glutathione (GSH) observed in the majority of tumors combined with its role in tumorigenesis, inflammation response and tumor microenvironment properties are explored for different therapy approaches. *Direct approaches* are aimed at blocking synthesis of GSH leading to ferroptosis of cells or more efficient chemotherapy; *indirect methods* target inflammation response or tumor microenvironment in order to make tumors more susceptible to the response of the immune system as well as immunotherapies; *elevated levels of GSH* and possibly other active molecules in tumor cells are used in the prodrug approach, where the drug is only activated after entering tumor cells utilizing GSH in the activation reaction or neutralizing it following activation with other factors.

**Figure 2 biomolecules-10-01429-f002:**
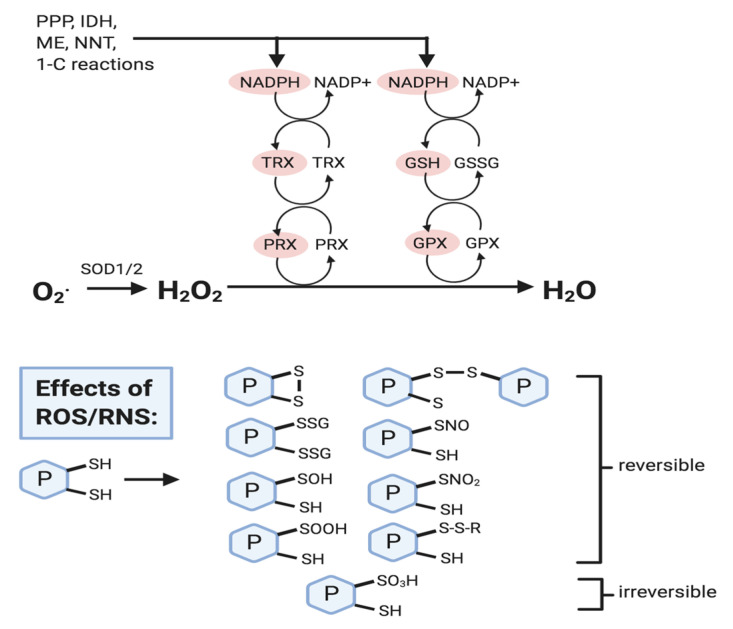
Reactive oxygen species (ROS) removal and effects. NADPH levels are crucial in ROS removal mechanisms and is produced through the pentose phosphate pathway (PPP), isocitrate dehydrogenase (IDH), malic enzyme (ME), nicotinamide nucleotide transhydrogenase (NNT), and one-carbon reactions (e.g., tetrahydrofolate metabolism). NADPH reduces oxidized forms of thioredoxin (TRX) and glutathione (GSSG), and these reduced forms (shown in red) then activate peroxiredoxin (PRX) and glutathione peroxidase (GPX) so that they can reduce H_2_O_2_ to water (H_2_O). ROS (mainly H_2_O_2_) and RNS (e.g., nitric oxide) can cause the oxidative modification of protein cysteine thiol residues, most of which are reversible, and serve to protect them from oxidative damage, and to modify protein activity. Refer to text for details.

**Figure 3 biomolecules-10-01429-f003:**
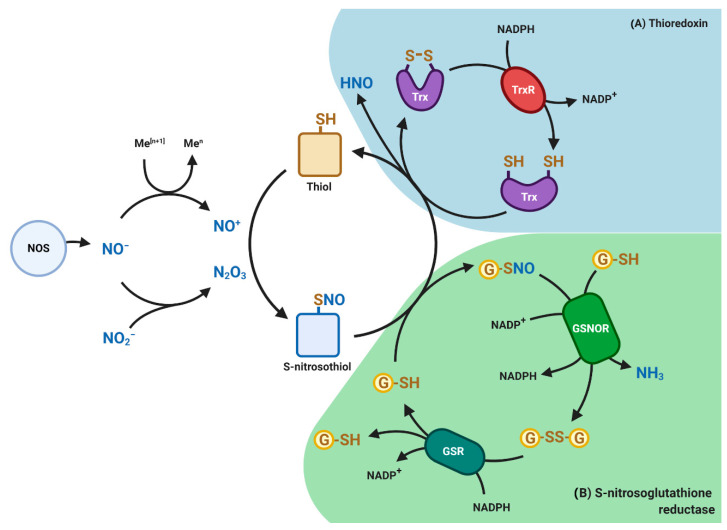
Redox-based post-translational modifications mediated by nitric oxide (NO). Nitric oxide synthases (NOS) generates NO gas which reacts with thiol-containing molecules such as proteins and glutathione (GSH) to form S-nitrosothiols or S-nitrosoglutatione (GSNO), respectively. Shown are (**A**) Thioredoxin (Trx) pathway (blue) and (**B**) S-nitrosoglutathione reductase (GSNOR) pathway (green). GSNOR and TR catalyzes the denitrosation of protein nitrosothiols and GSNO. By reducing GSNO to GSSG and NH_3_, GSNOR indirectly controls the levels of S-nitrosothiols and associated signal transduction. Glutathione (GSSG) is reduced back to GSH by glutathione reductase (GSR) at the expense of NADPH. Similarly, Trx mediates denitrosation of protein nitrosothiols by releasing nitroxyl (HNO) group to form reduced thioredoxin at the expense of NADPH. HNO can diffuse and react in the surrounding environment. An imbalance in NO signaling can promote tumorigenesis and tumor progression.

**Figure 4 biomolecules-10-01429-f004:**
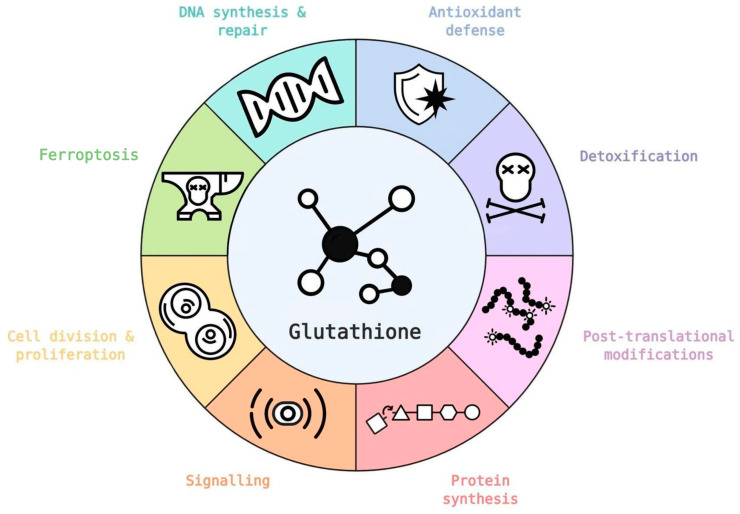
Glutathione has a number of critical roles in healthy and tumor cells, in both cases supporting cell metabolism and survival. Several glutathione functionalities are directly or indirectly regulated by oncogenes (icons based on database [50]). Refer to text for further details.

**Figure 5 biomolecules-10-01429-f005:**
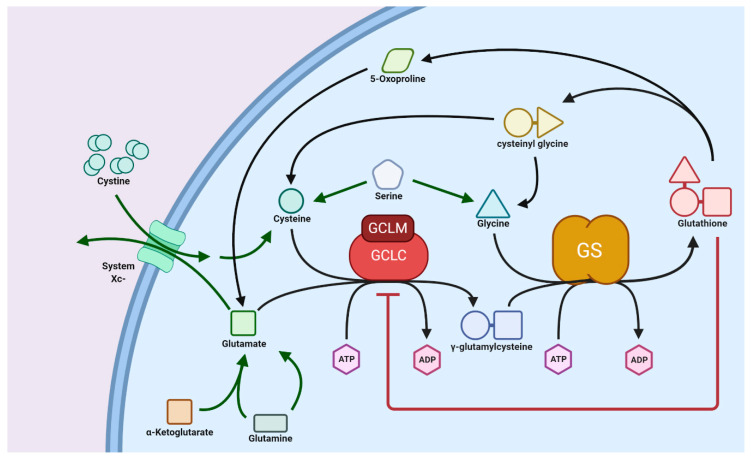
Biosynthesis pathway of glutathione (GSH) from amino acid precursors. GSH is synthesized in two consecutive ATP-dependent reactions within the cytosol coupled with GSH degradation to its constituent amino acids via 5-oxoproline and cysteinyl glycine, which forms the GSH cycle. Precursor sources are indicated by green arrows. GSH synthesis is regulated through inhibition of GCLC by GSH. GCLC, glutathione cysteine ligase (GCL) catalytic subunit; GCLM, glutathione cysteine ligase (GCL) modifier subunit; GS, glutathione synthase. Further explanation of synthesis processes is provided in the text.

**Figure 6 biomolecules-10-01429-f006:**
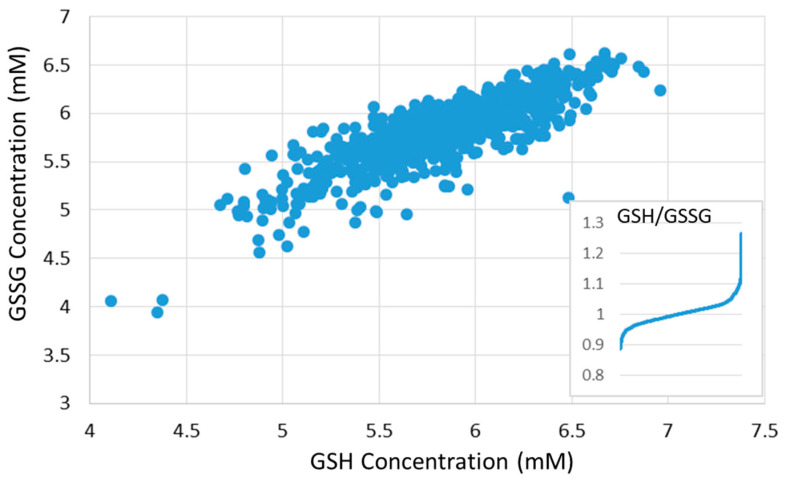
Relative GSH and GSSG concentrations vary slightly across cancer cell lines (data presented by Li et al., [65] and obtained from https://portals.broadinstitute.org/ccle). Figure shows concentration of GSSG as a function of the concentration of GSH in the same cell line (concentrations shown in mM). Insert plot shows differences in the ratio of GSH/GSSG across 928 cell lines.

**Figure 7 biomolecules-10-01429-f007:**
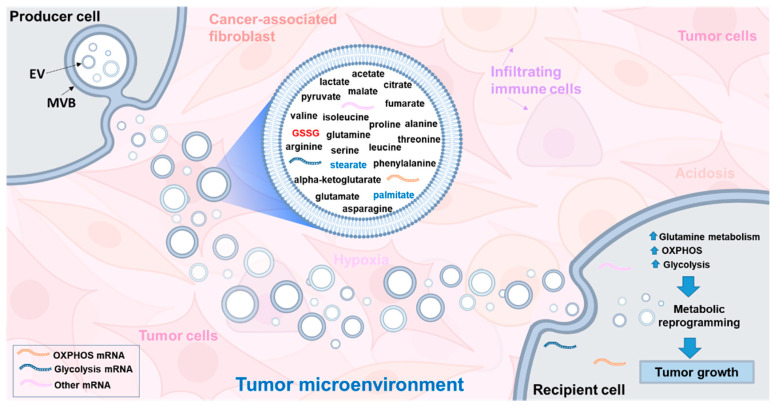
Induction of metabolic reprogramming in recipient cells by internalization of producer cell-derived exosomes. Producer cells (e.g., cancer-associated fibroblasts) contain high levels of tricarboxylic acid (TCA) cycle metabolites, amino acids and lipids that can fuel the metabolic activity of recipient tumor cells contributing to tumor progression. Producer cells (e.g., glioma cells) derived exosomes can also harbor mRNA encoding ribosomal, oxidative phosphorylation and glycolytic proteins that can reprogram glycolysis and oxidative phosphorylation in recipient cells resulting in oncogenic reprogramming and transformation. Intraexosomal cargoes, such as metabolites, amino acids and mRNA are based on the studies of Cuperlovic-Culf et al., [156] Zhao et al., [149] and Zheng et al., [150]. EV, extracellular vesicles, such as exosomes; MVB, multivesicular body; OXPHOS, oxidative phosphorylation. Refer to text for more details.

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
