# Peer review of "Role of Glutathione in Cancer: From Mechanisms to Therapies"

_biomolecules, 2020, doi:10.3390/biom10101429_

Round 1

Reviewer 1 Report

The manuscript is well written and covers a significant literature in glutathione metabolism and how it is modulated in various cancer studies.  I enjoyed reading it and I am sure it will be well received in the scientific community. My only comment involves minor grammatical corrections. Authors should go through the manuscript carefully. For example the sentence starting on line 737 “One of the major advantages to detailed kinetic models...” should read as  “One of the major advantages of detailed kinetic models”

Author Response

Thank you very much for your positive comments and interest in our manuscript. We have corrected the sentence as suggested and also read through the manuscript in great detail and corrected any grammatical and typographical errors.

Reviewer 2 Report

In this review the authors explore the role of glutathione in cancer taking into consideration many different aspects of GSH metabolism and function. The review is comprehensive and well-organized. Maybe addition of a short paragraph addressing more specifically the connection of glutathione with iron and Fe-S metabolism, ferroptosis and hypoxia would complete the picture. There are only a few very minor points that the authors should address.

Minor points and typos:

1) line 44: NADH not NADH2

2) lines 322, 699 and 778: correct paragraph 3.2 not 3.3 and also 8. Examples of GSH-related tumor therapies and 9. Conclusions 

3) in fig 2 it might be easier to add subscripts 'ox' and 'red' to better evidence the TRX, PRX and GPX oxidized and reduced forms

4) labels to axes are missing in fig. 6

Author Response

Thank you very much for your positive comments and interest in our manuscript. We have made all the recommended corrections including figure label and section numbers. Also, we have added a paragraph and another 2 references further discussing link between glutathione and ferroptosis.